# The Influence of Sous Vide Parameters on Nutritional Characteristics and Safety of Pikeperch Fillets

**DOI:** 10.3390/foods11111605

**Published:** 2022-05-29

**Authors:** Monika Modzelewska-Kapituła, Renata Pietrzak-Fiećko, Arkadiusz Zakrzewski, Adam Więk

**Affiliations:** 1Department of Meat Technology and Chemistry, Faculty of Food Sciences, University of Warmia and Mazury in Olsztyn, 10-719 Olsztyn, Poland; adam.wiek@uwm.edu.pl; 2Department of Commodities and Food Analysis, Faculty of Food Sciences, University of Warmia and Mazury in Olsztyn, 10-719 Olsztyn, Poland; renap@uwm.edu.pl; 3Department of Industrial and Food Microbiology, Faculty of Food Sciences, University of Warmia and Mazury in Olsztyn, 10-719 Olsztyn, Poland; arkadiusz.zakrzewski@uwm.edu.pl

**Keywords:** anserine, carnosine, cooking, fatty acids, microbial quality, *Sander lucioperca*

## Abstract

The aim of the study was to investigate the influence of temperature and time combination on the quality of pikeperch fillets and to propose settings which allow high nutritional quality fish fillets to be obtained. The material for the study consisted of 24 farmed pikeperch (*Sander lucioperca*) fillets, which were evaluated raw and after sous vide (SV) cooking proceeded at 65 °C for 45 min (SV65), 75 °C for 20 min (SV75), and 90 °C for 10 min (SV90). The chemical composition was affected by SV procedure; SV90 was similar to raw samples in terms of moisture, protein, and fat content, whereas SV65 differed the most. Carnosine contents decreased in all SV samples compared with raw ones, and anserine only decreased in SV90. There were no differences in terms of fatty acid composition (% of total) between SV and raw samples. In SV75 and SV90 total viable counts, *Enterobacteriaceae*, *Enterococcus* sp., and *Staphylococcus* sp. were reduced below a detection level but not in SV65. These samples also showed a better sensory quality than SV65. Therefore, SV75 and SV90 might be recommended for pikeperch fillets preparation, taking into account safety and nutritional aspects.

## 1. Introduction

Sous vide (SV) is a thermal treatment method which involves cooking of vacuum-packed foods under controlled conditions of temperature and time to gain a satisfactory sensory quality and to prolong products’ shelf-life [1]. The method applied to fish and fish-based products has some advantages compared with conventional cook–chill processes, such as limiting protein and lipids heat damage, enhancing collagen gelatinization, and lowering cooking losses and losses of heat-labile nutritional compounds [2]. The process might be conducted within a wide range of temperature from 50 °C to 95 °C [1,2,3,4], over short (i.e., 1.04 min at 90 °C in rainbow trout [3]) or long periods (up to 2 days for tough beef cuts at the temperature from 55 °C to 60 °C) [1]. To preserve SV products, rapid cooling and storage at a low temperature are required [5]. In the case of SV fish products stored below 3 °C, a shelf-life of even 4 to 9 weeks can be achieved [2]. In the scientific literature, different settings for preparing fish products using SV are proposed for different fish species, such as 80 °C for 45 min for salmon [2], 65 °C for 46 min for seabream [6], or 90 °C for 1.0 min and 3.3 min for rainbow trout [3]. However, to the authors’ best knowledge, no study has been conducted so far using pikeperch fillets. According to González-Fandos et al. [3], the application of too high of a temperature to fish might decrease the sensory quality of SV products; therefore, a heat treatment ranging from 60 °C to 80 °C for 20 min to 40 min had been suggested [3]. A temperature of about 70 °C might be suitable for obtaining a desirable texture of fish tissue, especially those with a higher collagen content [2]. However, there is still concern about the safety of sous vide fish cooked at a low temperature for a short time, which might not be sufficient to inactivate microbiota. Moreover, different fish species differ in tissue composition, such as fat and connective tissue contents. These components affect the effectiveness of thermal treatments in relation to the reduction of microbiota and shape the eating quality of final products. According to our best knowledge, there is a shortage of information about the effect of SV parameters on the quality of pikeperch. Pikeperch (*Sander lucioperca*) is a freshwater fish species, which is highly appreciated by consumers due to its white, tasty muscle tissue and relatively low fat content. From a nutritional perspective, consumers are encouraged to consume fish due to a high content of protein, which is highly digestible, polyunsaturated fatty acids (PUFA), including those from the n-3 group, such as eicosapentaenoic (EPA, C20:5, n-3), docosapentaenoic (DPA, C22:5, n-3), and docosahexaenoic (C22:6, n-3), and mineral compounds and vitamins [7]. Apart from those nutritional effects, the consumption of fish makes the risk of carcinogenesis lower [7,8].

Other bioactive compounds present in the muscle tissue of animals are histidine dipeptides, such as carnosine and anserine [9,10]. Carnosine is made of l-histidine and beta-alanine, whereas anserine is produced from carnosine as a result of metabolic changes involving its methylation [11]. Carnosine is identified in the tissues where metabolic processes take place most effectively—in the central nervous system, liver, kidneys, stomach, and skeletal muscles [11,12]. Anserine and carnosine show chelating properties against heavy metals, such as cobalt, zinc, iron, and copper. Through the chelation reactions of copper and zinc ions, carnosine regulates the level of zinc in the central nervous system, thus demonstrating a neuroprotective effect due to the fact that disturbances in zinc concentration in the central nervous system are associated with an increased risk of neurological diseases [13,14]. Carnosine and anserine exhibit strong antioxidant properties comparable to synthetic antioxidants such as butylhydroxytoluene (BHT) [15], inter alia, by inactivating free radical reaction products, deactivating peroxide and hydroxyl radicals, and blocking singlet oxygen, proxynitrile radicals, and chlorates [16,17]. The activity of carnosine as an antioxidant in the neutralization of free oxygen radicals does not inhibit its signalling and regulatory functions [18]. The anti-glycation and anti-carbonylation properties of carnosine and anserine have also been demonstrated [17,19]. The presence of carnosine and anserine in the protein–fat matrix contributes to the reduction of the formation of advanced products of lipid oxidation and advanced glycosylation end products [17,19]. Carnosine and anserine are identified in beef [20,21], pork [22], and poultry [11,23] and some fish and seafood [17]. Their content in the tissues is differentiated by the animal species, race, sex, vital activity of the muscle, and living conditions [11,17]. According to our best knowledge, no information on the concentration of anserine and carnosine in raw and sous vide pikeperch existed in the literature; thus, the present study fills the gap.

Apart from a high nutritional value and health-promoting properties, fish are challenging food products to prepare, especially when they are purchased unprepared for a thermal treatment (with guts and scales); therefore, ready-to-eat fish dishes are attractive to consumers [6]. Fish SV products will fill the gap in the market for ready-to-eat fish products after short reheating. On the other hand, rapid cooking methods are appreciated in industrial and catering practices; therefore, three different temperature and time combinations were investigated in this work. The aim of the study was to investigate the combined effect of time and temperature of sous vide treatment on composition, including fatty acids composition, anserine and carnosine contents, and sensory quality of pikeperch. The study was also aimed at setting guidelines for preparing this fish using the sous vide method that allow high nutritional quality fish fillets to be obtained.

## 2. Materials and Methods

### 2.1. Fish and Experiment Design

The material for the study consisted of 12 farmed pikeperch (*Sander lucioperca*) fillets (*n* = 24). The fillets were kindly provided by Stanisław Sakowicz Inland Fisheries Institute in Olsztyn (Poland). Briefly, the fish were produced in a recirculating aquaculture system with formulated feed. They were harvested at the age 2+, average body weight 1200 g, and average total length 50 cm. After the harvest, the pikeperch were killed and kept on ice for 6 h. In the study, skinned fillets were used, which were obtained after gutting, de-heading, fin removal, filleting, and skinning the fish. Fillets obtained from each fish were divided into four major samples: raw, assigned to analyses conducted on uncooked tissue (chemical composition, including fatty acids and anserine and carnosine contents), and three subjected to sous vide treatment at 65 °C for 45 min (SV65), 75 °C for 20 min (SV75), and 90 °C for 10 min (SV90). Each sample assigned to cooking was divided into two subsamples, vacuum-packed (pressure 25 mbar) individually in plastic pouches suitable for sous vide treatment (70 μm thick, 160 mm × 230 mm, PAPE nylon/polyethylene, Hendi Food Service Equipment, Rhenen, The Netherlands), and subjected to heating in a sous vide device (Hendi GN2/3, 400 W, Hendi Food Service Equipment, Rhenen, The Netherlands) on the same occasion. One subsample was used for microbiological analyses and sensory evaluations, whereas the other was weighted before and after cooking to determine cooking losses and then used to determine the chemical composition (Figure 1). The average thickness of sous vide samples was 2.2 cm and weight about 100 g.

### 2.2. Chemical Composition

Before chemical analyses, raw and sous vide fillets were individually ground twice through a size 4 mm mesh. Moisture content was determined using oven drying method at temperature 103 ± 2 °C [24]. Fat was extracted from the dried samples using ethyl ether according to Soxhlet method [25], crude protein content was determined with the use of the Kjeldahl method with 6.25 multiplier, and ash content was determined by sample mineralization at 550–600 °C [26]. The analyses were performed in duplicate.

### 2.3. Fatty Acid Composition

To determine fatty acid composition, the method described in Modzelewska-Kapituła et al. [27] was used. Muscle lipids were cold extracted with chloroform/methanol (2:1 *v*/*v*) [28]. Then, fatty acids were methylated with a chloroform/methanol/sulfuric acid (100:100:1) mixture [29] and separated using an Agilent Technologies 7890A gas chromatograph (Agilent Technologies, Inc., Santa Clara, CA, USA) with a flame-ionization detector (FID) (30 m, 0.32 mm internal diameter fused silica capillary column (matrix active group: poly(ethylene glycol)phase, Supelco, Bellefonte, PA, USA)). The liquid phase was Supelcowax 10, and the film thickness was 0.25 μm. The conditions of separation were as follows: carrier gas, helium; flow rate, 1 mL min; detector temperature, 250 °C; injector temperature, 230 °C; column at temperature, 195 °C. The fatty acids were identified by comparing retention times with standards from Supelco (Bellefonte, PA, USA). The fatty acid content was presented as the relative percentage (% total fatty acids) and concentration (mg/100 g), which was calculated based on the fat content and the coefficient for lean fish (0.70) [30].

### 2.4. Carnosine and Anserine Contents Determination

Carnosine and anserine were extracted using the method described in Modzelewska-Kapituła at al. [20]. The carnosine content of meat was determined by high-performance liquid chromatography (HPLC), derivatizing the extracts with phthalaldehyde (OPA, Sigma-Aldrich Chemie GmbH, Stainheim, Germany) working solution. The derivatized samples were analysed on the Thermo Scientific ACCELA chromatograph using Thermo Scientific ChromQuest 5.0 software (Thermo Fisher Scientific, Waltham, USA). The separation was carried out on a Venusil SCX column, 3 μm, 4.6 × 150 mm (Agela Technologies, Tianjin, China) under isocratic elution conditions at an eluent flow velocity of 750 μL/min at 25 °C. The eluent consisted of 0.5 M acetate buffer adjusted to pH 4.6, acetonitrile, and methanol in a volume ratio of 85:5:10. Histidine dipeptides were detected with the ACCELA PDA detector at λ = 332 nm. The calibration curve was determined by the external standard method for carnosine and anserine (Sigma-Aldrich Inc., St Louis, MO, USA). In the concentration range for each of them, 0.16–1.50 µg/25 µL, the determination coefficient R2 was greater than or equal to 0.99. The content of carnosine and anserine in meat samples was determined using a component of the ChromQuest 5.0 Concentration Calculator program. Two replicates were prepared from each muscle, and two sub-samples of each extract were analysed by HPLC.

### 2.5. Microbiological Analyses

To conduct microbial analyses, homogenates were prepared using 10 g of pikeperch tissue. The tissue was aseptically obtained by cutting slices from the dorsal, ventral, and tail area, weighted and transferred to 90 mL sterile saline (0.85% NaCl) and homogenized with a stomacher (Masticator Homogenizer Silver, IUL S.A., Spain). The homogenized sample was serially diluted using the same diluent (1:10 (*v*/*v*)). For microbial enumeration, 0.1 mL of each dilution was spread onto the surface of the sterile dry media. Total *Enterobacteriaceae* were enumerated using VRBL agar, and the plates were incubated for 24 h at 37 °C, whereas *Staphylococcus* sp. were enumerated using Baird-Parker agar, *Listeria monocytogenes* using ALOA agar, and *Enterococcus* sp. using Slanetz–Bartley agar incubated for 48 h at 37 °C. Total viable counts (TVC) were enumerated by inoculating 1.0 mL of the sample in a Petri dish and adding 20 mL of liquid (50 °C) nutrient agar. After setting, samples were incubated for 72 h at 30 °C. Microbial data are reported as numbers of colony forming units per gram (CFU/g). All the media used in the study originated from Merck (Darmstadt, Germany).

### 2.6. Sensory Analysis

The analysis was carried out according to Żmijewski et al. [31]. Briefly, just after the termination of sous vide treatment, each fillet was cut on approx. 2.0 cm × 2.0 cm samples, which were evaluated by a sensory panel composed of 6 assessors (each assessor evaluated a sample from each fish, cut from the same fillet part). The sensory evaluation was carried out in four separate sessions (6 samples per session). The assessors performing the sensory assessment met the minimum sensory competence required in the current calibrating tests; they were selected, trained, tested, and monitored according to ISO 8586-1 guidelines [32]. The assessors were experienced in sensory evaluation and were acquainted with the accepted system of sensory assessment. Colour, aroma, texture, juiciness, and taste were determined on a 5-point scale (Table 1).

### 2.7. Statistical Analysis

To compare the mean values, the normal distribution of data (Shapiro–Wilk’s test) and variance homogeneity (Leven’s test) were tested. These variables, which showed a normal distribution and homogeneity of variance, were then subjected to variance analysis and Tukey’s honest significant difference (HSD) test to determine differences between treatments. Those variables which did not fulfil the assumption of normal distribution and variance homogeneity (carnosine and anserine contents) and sensory analysis results were compared using a non-parametric Kruskal–Wallis test. All the calculations were conducted in Statistica 13.3 software (Tibco Software Inc., Palo Alto, CA, USA).

## 3. Results and Discussion

### 3.1. Proximate Composition and Carnosine and Anserine Contents

Both raw and sous vide cooked pikeperch fillets had low amounts of fat (Table 2), which supports the image of this fish as a lean one. Due to the fat content being below 3%, it can be labelled as a low-fat product in European Union countries (regulation (EC) no. 1924/2006 of the European Parliament [33]). The content of protein in raw and sous vide fillets exceeded 20%, which is a high value and beneficial from a nutritional perspective.

Generally, sous vide cooking itself affected the chemical composition, and significant differences were noted between raw and cooked fillets (Table 2). However, the combination of a temperature height and time of sous vide also affected the results. In SV65, the highest cooking loss was noted (compared with SV75 and SV90), which resulted in a significant moisture content decrease and an increase in protein, fat, and ash contents compared with raw samples. A decrease in moisture content and an increase in protein, fat, and ash contents were noted also in SV75 compared with a raw tissue but not in SV90, in which only an increase in ash content was noted. Çağlak et al. [34], who subjected pikeperch fillets to sous vide at 60 °C, 70 °C, and 90 °C for 10 min, found no differences in moisture and fat contents between treatments, whereas protein content increased along with a temperature increase, and ash decreased in the samples cooked at 70 °C and 90 °C compared with 60 °C. Changes in the chemical composition of fish muscle tissue related to a treatment’s temperature resulted from the cooking losses as well as changes in fat components such as polymerization and oxidation of the triglycerides and release of neutral lipids [35].

Based on the literature data, it should be stated that the content of histidine dipeptides in fish tissue is very diverse. Fish species with a low content of histidine dipeptides include, among others, herring, sardines, salt, carp, cod, and trout. A rich source of compounds from this group are, in turn, tuna, salmon, and croakers [17]. Importantly, in the total number of identified compounds from the group of histidine dipeptides in individual fish species, the share of one of them is dominant, usually anserine and less frequently carnosine [17]. The total content of carnosine and anserine determined in this study was over 54 mg/100 g of raw pikeperch tissue, of which almost 73% was anserine. The average content of anserine in raw pikeperch (39.42 mg/kg) demonstrated in our research corresponds to the content of this compound determined by Jones et al. [36] for mackerel (2.0 mmol/kg). At the same time, the content of carnosine shown in our research (14.61 mg/kg) was similar to the content determined by Jones et al. [36] for rainbow trout tissue (0.5 mmol/kg). Taking into account the literature data showing a significantly higher content of carnosine and anserine in pork, poultry, and, in particular, beef, pikeperch meat should be considered a material low in histidine dipeptides.

Sous vide cooking reduced the concentration of carnosine regardless of the sous vide parameters used (Table 2). However, in SV65 the concentration of carnosine was twice as much as in SV75 and SV90 samples, but the difference was not of statistical significance. In terms of anserine, there were no differences between raw, SV65, and SV75 samples; however, in SV90 a significantly lower concentration of the compound was noted. On the other hand, there were no significant differences between SV75 and SV90 in anserine contents, which indicates that consumption of these fillets would deliver a similar amount of this bioactive compound.

The initial content of histidine dipeptides identified in fish tissue may change as a result of the heat processes used. The presented research results indicate that the reduction of carnosine content compared to the raw material was nearly 70% with SV65 treatment and over 80% with SV75 and SV90. The reduction of the initial amount of anserine in the pikeperch as a result of the applied SV treatment was significantly lower and amounted to a maximum of 16% for the SV90. This relationship is consistent with the findings of Peiretti et al. [21] after analysing the effect of various methods of heat treatment on the content of dipeptides in beef and turkey products. They showed that the final content of dipeptides in the products may be lower than the initial content by up to 70%, with the greatest losses of carnosine and anserine found in cooked samples. Higher losses of dipeptides due to heat treatment were found in beef samples than in turkey samples. The main reasons for the loss of dipeptides during heat treatment are the high solubility of dipeptides in water, in particular carnosine, heat leakage with aggregate peptides, and the presence of other nutrients, such as fat, limiting the possibility of protein and dipeptide aggregation in the structure of heated products [21,37]. Pereira-Lima et al. [38], analysing the content of carnosine and anserine in model beef broths, differentiated by temperature and heating time, indicated the heating temperature as a factor significantly differentiating the effectiveness of the extraction of dipeptides from meat. The increase in temperature in the range from 55 °C to 100 °C resulted in an increase in the content of dipeptides in the broths. The authors justify the demonstrated relationship between the content of anserine and carnosine in broths and the extraction conditions, the solubility of anserine and carnosine and their release from tissues being a result of structural changes caused by heating.

### 3.2. Fatty Acids Proportion and Concentration

The fatty acid composition (% of total fatty acids) is presented in Table 3. The fatty acids composition of pikeperch tissue used in this study differs from that reported by Jankowska et al. [39] for farmed pikeperch, where the most abundant were PUFA (41% of total fatty acids vs. 28% in this study), followed by MUFA (monounsaturated fatty acids, 31.4% vs. 52%) and SFA (saturated fatty acids, 27.5% vs. 20%), which might be explained by different rearing conditions of fish. This also highlights that the most important factor shaping the quality of sous vide fish products in terms of fatty acid composition is the profile of fatty acid in raw fish muscle tissue. The n-6/n-3 ratio was below 1.0 in all samples. This low n-6/n-3 ratio indicates the potential cardio-protective properties of pikeperch fatty acids, even after sous vide cooking up to 90 °C. The lower the n-6/n-3 ratio, the better the health benefits resulting from the consumption of the products associated with cardiovascular disease and other chronic disorders risk reduction [35].

Interestingly, sous vide treatment, irrespectively of the temperature and time combination used, did not change fatty acid composition, and no differences between samples were noted. This indicates that the sous vide treatment proceeded under conditions used in this study preserved all fatty acids. Contrasting results were reported by Redfern et al. [35], who subjected brined and un-brined salmon to sous vide conducted at 52 °C, 65 °C, and 80 °C and noted that the PUFA content in the un-brined salmon cooked at 80 °C for 15 min was significantly reduced compared with other sous vide cooked samples and raw salmon. Redfern et al. [35] concluded that higher temperatures of sous vide caused a modification of the double bonds of the PUFA, producing more saturated forms such as the MUFA and SFA. Differences between the results obtained in the study of Redfern et al. [35] and this study might be explained by a different fish species used. According to the authors’ best knowledge, no data concerning the effect of sous vide treatment on pikeperch fatty acid composition already exist in literature; thus, a comparison of the presented data with other reports is difficult. On the other hand, comparing the results of the present study to those conducted on other fish species might be misleading to some extent due to a different content of antioxidants in the muscle tissue of different fish species, which might have a key role against the lipid oxidation and changes in fatty acid composition as a result of a thermal process [40].

Due to an increase in fat content as a result of fillets heating, some differences in fatty acid contents (mg/100 g) were noted between raw and cooked samples. These changes were the most pronounced in SV65, in which an increase in SFA, MUFA, PUFA, n-3 (Figure 2), EPA, DPA, (Figure 3), and n-6 concentrations was noted compared with raw fillets (Figure 2). However, SV65 did not differ from SV75 and SV90, which indicates that all of combinations studied produced sous vide fillets with a similar nutritional value in terms of fatty acids contents. Interestingly, no differences between raw and sous vide cooked samples were noted in terms of DHA acid concentration (*p* > 0.05) (Figure 3). The sous vide technique used in the study turn out to be non-destructive towards PUFA in pikeperch fat. In contrast, traditional thermal treatment methods, i.e., pan or deep-frying and steaming, have been found to degrade the n-3 PUFA in fish muscle tissue through their oxidation [41].

### 3.3. Sensory Quality

Sous vide fillets were assessed in terms of sensory quality using a one-to-five scale (Table 1), and results are presented in Figure 4. Generally, all samples gained high average notes, which indicates their good eating quality. Sous vide samples did not differ (*p* > 0.05) in terms of colour, juiciness, and taste. In terms of aroma, SV75 and SV90 received higher notes than SV65 samples, which indicates that SV75 and SV90 were characterized by a typical, clearly perceptible aroma. There was a significant (*p* < 0.05) difference between the texture of samples—SV90 scored higher than SV65, whereas SV75 samples were similar to both SV65 and SV90. Although SV65 showed a higher cooking loss than SV75 and SV90, the difference was not big enough to produce a difference in fillet juiciness.

Results obtained in the study resemble those reported by Cai et al. [42], who proceeded a sous vide treatment on Russian sturgeon fillets at 40 °C, 50 °C, and 60 °C for 10 min. They found only slight differences in sensory quality between treatments, which were limited to a decrease in colour acceptability in those cooked at 60 °C. In the present study, all treatments were processed over the denaturation temperature for myoglobin (60 °C), which is a main component affecting the colour of muscle tissue. The colour of cooked muscle tissue, and its texture as well, is affected also by changes in muscle proteins, which started to take place over 40 °C [43]. In fish muscle tissue, collagen is converted into gelatine at temperatures ranging from 46°C to 49 °C [44]. The shrinkage of muscle fibre occurs between 40 °C and 50 °C (caused by myofibrillar protein denaturation) and between 65 °C and 70 °C (collagen filers shrinkage) [43]. However, the scope of denaturation, aggregation, and degradation of myofibrillar, sarcoplasmic, and connective tissue proteins depends on the combination of time and temperature during the heat treatment [45], which explains differences between treatments in this study. The higher the temperature, the greater the shrinkage is—i.e., it was lower at 65 °C than at 75 °C and 85 °C in trout [46]. The development of a typical aroma for cooked muscle tissue starts over 70 °C [43]; therefore, the aroma of SV75 and SV90 was scored higher in this study.

### 3.4. Microbiological Quality

Temperatures of 75 °C and 90 °C were effective in reducing TVC, *Enterobacteriaceae*, *Enterococcus* sp., and *Staphylococcus* sp. below the detection level of 10 CFU/g. In SV65 samples, a three-fold decrease in TVC and *Enterobacteriaceae* was noted, whereas in terms of *Staphylococcus* sp. only a one-fold reduction was noted. In these samples, only a slight reduction of *Enterococcus* sp. was noted, and the number remained at 10^2^ CFU/g (Table 4). A microbial reduction as a result of sous vide cooking was observed also by others and resulted in shelf-life extension. Çağlak et al. [34] reported that sous vide conducted at 60 °C, 70 °C, and 80 °C for 10 min caused a similar reduction in mesophilic and psychrotrophic bacteria counts on the next day after the treatment, irrespectively of the temperature used. However, during storage the counts started to increase, and the increase was more dynamic in samples processed at a lower temperature. It was concluded that sous vide treatment conducted at 60 °C, 70 °C, and 80 °C enabled the preservation of pikeperch fillets for 28, 35, and 56 days, respectively, when stored at 2 °C [34]. *Listeria monocytogenes* was not detected in raw pikeperch fillets in our study, so the reduction level was not determined. In the light of the current European Union regulations [47], the limit for *L. monocytogenes* in ready-to-eat foods able to support the growth of the pathogen, other than those intended for infants and for special medical purposes, is 100 CFU/g in products placed in the market during their whole shelf-life. According to The Rapid Alert System for Food and Feed (RASFF) in the EU, between 2010 and 2020, fish and fish products showed the highest number of reported contaminations by *L. monocytogenes* among all food product categories [48]. Therefore, the consumption of those products might be associated with an elevated risk of listeriosis. Dogruyol et al. [49] showed that a 90% reduction in *L. monocytogenes* counts (D-value) might be obtained after 1.49 min of sous vide cooking of salmon at 62.5 °C. Thus, it might be concluded that all temperature and time combinations used in the study would be effective in the inactivation of *L. monocytogenes* if the bacteria were present in a raw material.

## 4. Conclusions

From a nutritional perspective, using both SV75 and SV90 temperature and time combinations enables pikeperch fillets with preserved precious n-3 fatty acids to be obtained. These treatments did not differ in bioactive compounds (anserine and carnosine) concentration, microbiological quality, nor sensory quality. Taking into account a slightly lower cooking loss and a shorter time of thermal cooking, which would be appreciated by chefs, using 90 °C for 10 min might be recommended as a way to prepare sous vide pikeperch fillets. However, SV90 showed a lower anserine content compared with raw samples, whereas SV75 did not differ from raw tissue, which makes SV75 more nutritious. Moreover, based on literature data, the SV75 and SV90 pikeperch products should be suitable for prolonged storage. However, further study is needed to verify the safety of the fillets during storage and changes in fatty acid composition, with a special emphasis on PUFA.

## Figures and Tables

**Figure 1 foods-11-01605-f001:**
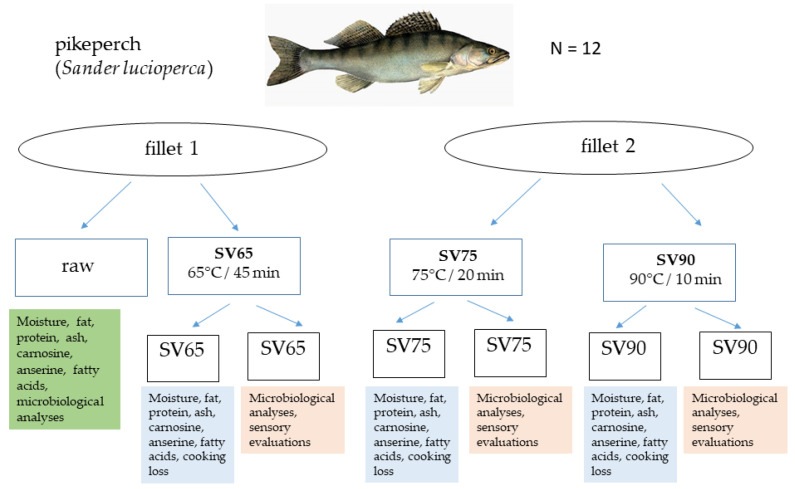
The experiment design.

**Figure 2 foods-11-01605-f002:**
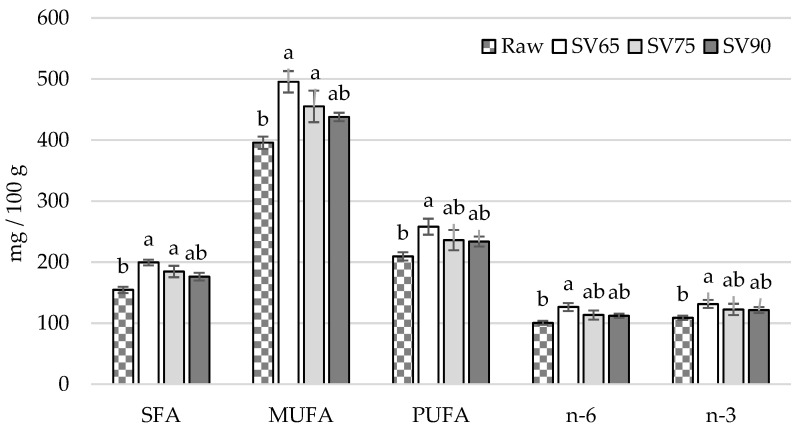
Comparison of fatty acids contents (mg/100 g) in pikeperch fillets: raw and after sous vide treatment proceeded at 65 °C for 45 min (SV65), 75 °C for 20 min (SV75), and 90 °C for 10 min (SV90); vertical bars refer to standard error of the mean; a,b—values denoted with different letters differ significantly at *p* < 0.05; SFA—saturated fatty acids; MUFA—monounsaturated fatty acids; PUFA—polyunsaturated fatty acids.

**Figure 3 foods-11-01605-f003:**
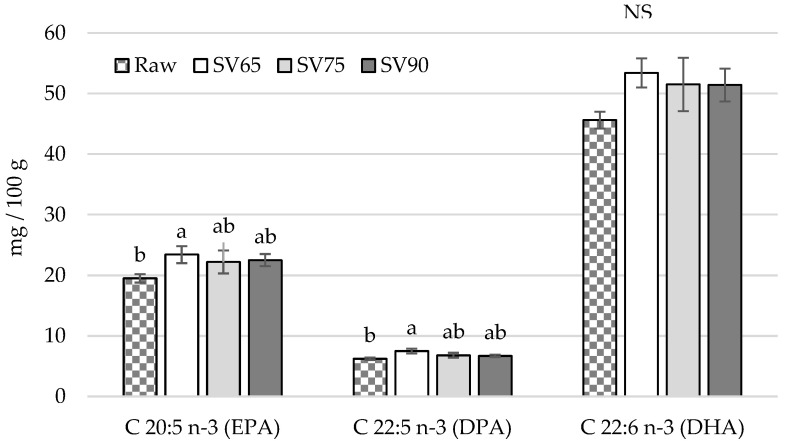
Comparison of fatty acids content (mg/100 g) in pikeperch fillets: raw and after sous vide treatment proceeded at 65 °C for 45 min (SV65), 75 °C for 20 min (SV75), and 90 °C for 10 min (SV90); vertical bars refer to standard error of the mean; a,b—values denoted with different letters differ significantly at *p* < 0.05; NS—no significant differences; EPA—eicosapentaenoic fatty acid; DPA—docosapentaenoic fatty acid; DHA—docosahexaenoic fatty acid.

**Figure 4 foods-11-01605-f004:**
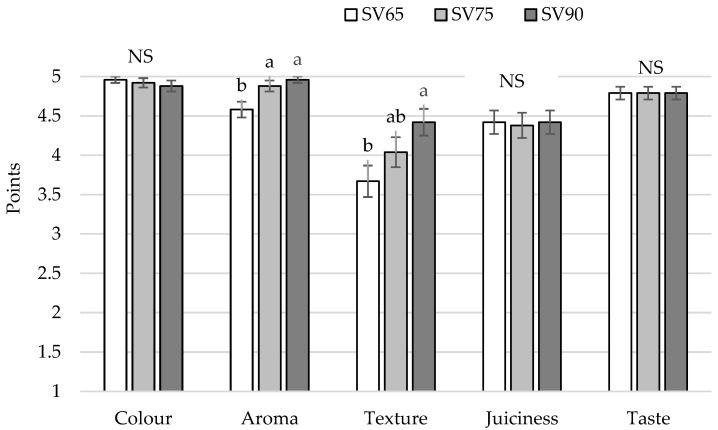
Comparison of sensory attributes of pikeperch after sous vide treatments proceeded at 65 °C for 45 min (SV65), 75 °C for 20 min (SV75), and 90 °C for 10 min (SV90); vertical bars refer to standard error of the mean; a,b—values denoted with different letters differ significantly at *p* < 0.05; NS—no significant differences.

**Table 1 foods-11-01605-t001:** Sensory analysis description and scores used in sous vide cooked pikeperch fillets evaluation.

Points	Colour	Aroma	Texture	Juiciness	Taste
5	Homogeneous, typical for cooked fish of certain species	Typical, clearly perceptible	Firm, particularly tender	Very juicy	Desirable, typical, intense, no foreign aftertastes
4	Homogeneous, darker than typical	Typical, poorly perceptible or very strong	Slightly too soft or too hard, very tender	Juicy	Desirable, typical, weak, no foreign aftertastes
3	Slightly inhomogeneous, too dark	Poorly perceptible, foreign	Semi-soft or hard, tender	Low juiciness	Typical with slight foreign aftertaste
2	Inhomogeneous	Perceptible, foreign	Soft or hard, fibrous	Dry	Clearly perceptible foreign aftertaste
1	Very inhomogeneous	Strong foreign	Very soft or very hard, fibrous	Very dry	Atypical, strong foreign aftertaste

**Table 2 foods-11-01605-t002:** Changes in the chemical composition during thermal treatment of pikeperch tissue (mean values and standard error of mean in brackets).

Attribute	Raw	SV65	SV75	SV90	*p* Value
Moisture (%)	76.73 ^a^ ± 0.27	74.40 ^c^ ± 0.23	75.33 ^bc^ ± 0.40	76.30 ^ab^ ± 0.33	***
Protein (%)	20.70 ^b^ ± 0.23	22.81 ^a^ ± 0.34	21.79 ^a^ ± 0.44	20.76 ^b^ ± 0.30	***
Fat (%)	1.09 ^c^ ± 0.03	1.36 ^a^ ± 0.05	1.25 ^ab^ ± 0.07	1.21 ^bc^ ± 0.03	***
Ash (%)	1.09 ^b^ ± 0.01	1.17 ^a^ ± 0.01	1.20 ^a^ ± 0.02	1.21 ^a^ ± 0.03	***
Carnosine (mg/100 g)	14.61 ^a^ ± 0.81	4.60 ^b^ ± 0.31	2.01 ^b^ ± 0.30	2.07 ^b^ ± 0.92	***
Anserine (mg/100 g)	39.42 ^a^ ± 0.97	44.54 ^a^ ± 2.60	38.03 ^ab^ ± 2.12	32.88 ^b^ ± 1.90	**
Cooking loss (%)	-	19.98 ^a^ ± 0.66	17.26 ^b^ ± 0.39	16.38 ^b^ ± 1.02	**

^a–c^—mean values with different superscript within age group differ significantly at *p* < 0.05; *** *p* < 0.001, ** *p* < 0.01; NS—no significant differences (*p* > 0.05); SV65—sous vide at 65 °C for 45 min; SV75—sous vide at 75 °C for 20 min; SV90—sous vide at 90 °C for 10 min.

**Table 3 foods-11-01605-t003:** Fatty acid composition (% of total) of raw and sous vide cooked pikeperch (mean values and standard error of mean in brackets).

Fatty Acid (%)	Raw	SV65	SV75	SV90
C 12:0	0.036 ± 0.001	0.034 ± 0.001	0.034 ± 0.001	0.039 ± 0.001
C 14:0	3.344 ± 0.035	3.397 ± 0.057	3.384 ± 0.059	3.354 ± 0.069
C 15:0	0.288 ± 0.003	0.295 ± 0.004	0.287 ± 0.005	0.293 ± 0.005
C 16:0	14.912 ± 0.511	15.493 ± 0.262	15.596 ± 0.436	15.381 ± 0.437
C 17:0	0.160 ± 0.001	0.160 ± 0.002	0.159 ± 0.002	0.160 ± 0.003
C 18:0	1.477 ± 0.012	1.427 ± 0.043	1.509 ± 0.053	1.405 ± 0.033
C 20:0	0.162 ± 0.003	0.155 ± 0.006	0.156 ± 0.005	0.159 ± 0.006
C 14:1	0.113 ± 0.005	0.124 ± 0.012	0.116 ± 0.010	0.116 ± 0.010
C 16:1	6.426 ± 0.160	6.150 ± 0.315	6.657 ± 0.216	6.354 ± 0.267
C 17:1	0.334 ± 0.003	0.348 ± 0.010	0.335 ± 0.004	0.327 ± 0.004
C 18:1 cis9	35.995 ± 0.231	36.371 ± 0.149	35.812 ± 0.312	35.524 ± 0.149
C 18:1 cis11	2.795 ± 0.042	2.842 ± 0.016	2.750 ± 0.035	2.866 ± 0.027
C 20:1 n-9	4.268 ± 0.098	4.091 ± 0.150	4.219 ± 0.100	4.354 ± 0.059
C22:1 n-11	1.833 ± 0.039	1.772 ± 0.079	1.765 ± 0.068	1.789 ± 0.061
C22:1 n-9	0.318 ± 0.006	0.304 ± 0.009	0.317 ± 0.016	0.319 ± 0.011
C 18:2 n-6	11.896 ± 0.166	11.942 ± 0.201	11.634 ± 0.233	11.917 ± 0.269
C 18:3 n-6	0.169 ± 0.004	0.173 ± 0.002	0.169 ± 0.007	0.157 ± 0.003
C 18:3 n-3	3.461 ± 0.044	3.475 ± 0.071	3.363 ± 0.057	3.406 ± 0.074
C 18:4 n-3	0.653 ± 0.007	0.649 ± 0.007	0.636 ± 0.010	0.637 ± 0.012
C 20:2 n-6	0.651 ± 0.007	0.651 ± 0.010	0.639 ± 0.013	0.645 ±0.013
C 20:3 n-6	0.152 ± 0.003	0.150 ± 0.002	0.152 ± 0.005	0.146 ± 0.002
C 20:4 n-6	0.254 ± 0.006	0.236 ± 0.005	0.250 ± 0.015	0.273 ± 0.008
C 20:3 n-3	0.323 ± 0.003	0.321 ± 0.007	0.317 ± 0.004	0.314 ± 0.006
C 20:4 n-3	0.497 ± 0.005	0.496 ± 0.009	0.482 ± 0.007	0.478 ± 0.007
C 20:5 n-3	2.561 ± 0.048	2.451 ± 0.063	2.519 ± 0.102	2.652 ± 0.069
C 22:5 n-6	0.105 ± 0.004	0.106 ± 0.004	0.105 ± 0.008	0.094 ± 0.006
C 22:5 n-3	0.812 ± 0.007	0.790 ± 0.019	0.781 ± 0.009	0.793 ± 0.019
C 22:6 n-3	6.004 ± 0.089	5.599 ± 0.046	5.857 ± 0.205	6.048 ± 0.177
SFA	20.38 ± 0.50	20.96 ± 0.31	21.12 ± 0.49	20.79 ± 0.51
MUFA	52.08 ± 0.30	52.00 ± 0.34	51.97 ± 0.20	51.65 ± 0.37
PUFA	27.54 ± 0.34	27.04 ± 0.42	26.90 ± 0.52	27.56 ± 0.49
n-3	14.31 ± 0.18	13.78 ± 0.22	13.95 ± 0.34	14.33 ± 0.27
n-6	13.23 ± 0.18	13.26 ± 0.21	12.95 ± 0.25	13.23 ± 0.28
n-6/n-3	0.93 ± 0.007	0.96 ± 0.004	0.93 ± 0.019	0.92 ± 0.017
n-3/n-6	1.08 ± 0.008	1.04 ± 0.005	1.08 ± 0.022	1.08 ± 0.019

SV65—sous vide at 65 °C for 45 min; SV75—sous vide at 75 °C for 20 min; SV90—sous vide at 90 °C for 10 min; NS—no significant differences between the treatments were noted (*p* > 0.05).

**Table 4 foods-11-01605-t004:** The effect of sous vide parameters on microbial quality of pikeperch fillets (the number of bacteria is expressed in CFU/g).

Bacteria Species/Group	Raw	SV65	SV75	SV90
TVC	9.08 × 10^5^	4.55 × 10^2^	<10	<10
*Enterobacteriaceae*	2.87 × 10^5^	2.00 × 10^2^	<100	<100
*Enterococcus* sp.	1.50 × 10^2^	1.00 × 10^2^	<100	<100
*Staphylococcus* sp.	1.05 × 10^3^	1.00 × 10^2^	<100	<100
*Listeria monocytogenes*	<100	<100	<100	<100

TVC—total viable counts; SV65—sous vide at 65 °C for 45 min; SV75—sous vide at 75 °C for 20 min; SV90—sous vide at 90 °C for 10 min; <100 CFU/g—number of bacteria below the detection limit.

## Data Availability

Datasets generated from the current experiment are available from the corresponding authors upon reasonable request.

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
