# Peer review of "The Influence of Sous Vide Parameters on Nutritional Characteristics and Safety of Pikeperch Fillets"

_foods, 2022, doi:10.3390/foods11111605_

Round 1

Reviewer 1 Report

Foods

foods-1727800

The influence of sous vide parameters on nutritional characteristics and safety of pikeperch fillets

Dear Editor,

The paper deals with the influence of sous vide cooking at different temperature and time combinations on the quality of pikeperch fillets. The paper has been well designed and written. Specific comments and questions;

  • Line 33: Only 16 h? To me, longer than 16h, check the literature!
  • “Moisture” should be “water” throughout the paper
  • Give more information about the GC conditions for fatty acid composition analysis
  • Line 154: “Microbial” should be “Microbiological”
  • Didn’t the authors determine the balenine content of their samples?
  • Give the method validation parameters for the chromatographic analyses
  • Line 311: Why?
  • Why is the statistical significance in the Table 3 and the Figure 1 different from each other?

Author Response

Dear Editor and Reviewers,

Thank you for valuable comments and suggestions and the possibility to improve our paper. Changes in text were marked with changes tracking mode. Below we enclosed the responses (R) for each Reviewers comments (C). 

Reviewer 1

General comment

C.The paper deals with the influence of sous vide cooking at different temperature and time combinations on the quality of pikeperch fillets. The paper has been well designed and written.

R. Thank you!

Specific comments and questions;

C1. Line 33: Only 16 h? To me, longer than 16h, check the literature!

R. Thank you for the comment. Indeed, in the paper by Diaz et al. (2011) a time of 16h is shown. However, in another source even two days are indicated for beef. The sentence in the manuscript was modified to show the range of temperature and time used in sous vide technique.

C2. “Moisture” should be “water” throughout the paper

R. In the scientific papers both expression are used. In AOAC standard the word “moisture” is used (“AOAC Official Method 950.46 Moisture in Meat”) therefore we would prefer to remain “moisture” in the manuscript.

C3. Give more information about the GC conditions for fatty acid composition analysis

R. Details were provided.

C4.Line 154: “Microbial” should be “Microbiological”

R. Corrected

C5. Didn’t the authors determine the balenine content of their samples?

R. The authors did not take into account the determination of the content of balenin in the conducted research. The Introduction section referring to histidine dipeptides was modified not to confuse the readers.

C6. Give the method validation parameters for the chromatographic analyses.

R. Completed in the methodological description.

C7. Line 311: Why?

R. Cooking caused changes in overall fat content in treatments which affected fatty acids concentrations expressed in mg/100g. The explanation was enclosed at the beginning of the sentence: “Due to an increase in fat content as a result of fillets heating…”.

C8. Why is the statistical significance in the Table 3 and the Figure 1 different from each other?

R. In Table 3 there is a relative proportion of FA (%FA of total) and in Fig. 1 results expressed as mg/100g. As it was shown in table 1, fat content in treatments differed which produced differences in mg/100g of particular FA and significant differences between treatments in fatty acids expressed in mg/100g. 

Reviewer 2 Report

The aim of the study was to study the influence of temperature and time combination on the quality of pikeperch fillets, during sous vide treatment.

The over all work is well structured.

However, I have some major concerns in the sensory analysis. Therefore, I would recommend reconsideration after major revisions.

Table 1- the terminology present in this table is difficult to be used in sensory analysis. Description associated to some parameters does not correspond to a specific state. Example: the terms “somewhat too soft”; “somewhat to hard”, are contradictory for the same value. Somewhat is not “concise term”, to describe a sample sensorial state. The descriptions in the points 4 and 3, should be concise as in 1, 2 and 5. The points 3 and 4 could be one point. The authors should have checked the used terms.

An example can be found in Boriolova 2016  https://doi.org/10.1016/j.meatsci.2016.03.024,  which is used as reference in the work of Cai 2021 (reference 42 in this work).

Due to this, I’m not confident with the results from the sensory analysis. I would recommend the repetition of the experiment with suitable terminology.

Also, English need extensive revision. Phrase structure have some deficiencies throughout the text.

Below follows some specific corrections.

Line 32-33: how long is short? 3min?

Line 38-40: where in the internet? Reference for this sentence

Line 40-42: from who? The reference is there but, the sentence is improved if the person/institution is nominated.

Line 44-45: rephrase please, (… which might not be sufficient…)

Line 47-49: with might have? Phrase structure is also strange. Rephrase please.

Line 48-52: if the information is highlighted it should be referenced.

Line 62-89: This paragraph is confusing to me. Please restructure it. Avoid alternation between histidine dipeptides and the specific amines (carnosine and anserine). Example Line 82-89, is better if connected with first sentence of line 62.

Line 241 – 242: Rephrase (… which indicates that consumption of these fillets were deliver a similar amount of 241 this bioactive compound.)

Line 390-391: remove

Author Response

Dear Editor and Reviewers,

Thank you for valuable comments and suggestions and the possibility to improve our paper. Changes in text were marked with a changes tracking mode. Below we enclosed the responses (R) for each Reviewers comments (C). 

 Reviewer 2

GC: The aim of the study was to study the influence of temperature and time combination on the quality of pikeperch fillets, during sous vide treatment. The overall work is well structured.

However, I have some major concerns in the sensory analysis. Therefore, I would recommend reconsideration after major revisions.

R. Thank you. We did our best to improve the quality of the manuscript. We appreciate all your comments and we are truly grateful for your contribution.

Specific comments

C1. Table 1- the terminology present in this table is difficult to be used in sensory analysis. Description associated to some parameters does not correspond to a specific state. Example: the terms “somewhat too soft”; “somewhat to hard”, are contradictory for the same value. Somewhat is not “concise term”, to describe a sample sensorial state. The descriptions in the points 4 and 3, should be concise as in 1, 2 and 5. The points 3 and 4 could be one point. The authors should have checked the used terms. An example can be found in Boriolova 2016  https://doi.org/10.1016/j.meatsci.2016.03.024,  which is used as reference in the work of Cai 2021 (reference 42 in this work). Due to this, I’m not confident with the results from the sensory analysis. I would recommend the repetition of the experiment with suitable terminology.

R. Thank you for the comment. Unfortunately, there is no possibility to repeat the sensory assessment for those samples. However, we examined results carefully and have revised descriptors. Some modifications were made in the Table 1, such as deleting the word “somewhat” and differentiating 4 and 3 for colour and texture. Those changes did not affect results and their description. We will bear in mind your comment in our future studies.

C2. Also, English need extensive revision. Phrase structure have some deficiencies throughout the text. Below follows some specific corrections.

R. English was revised and corrected.

C3. Line 32-33: how long is short? 3min?

R. Details were added in this sentence, providing parameters for short and long sous-vide.

C4. Line 38-40: where in the internet? Reference for this sentence

R. Due to the fact that the sentence referred to non-scientific source we decided to delete it.

C5. Line 40-42: from who? The reference is there but, the sentence is improved if the person/institution is nominated.

R. The sentence was rephrased, indicating according to whom.

C6. Line 44-45: rephrase please, (… which might not be sufficient…)

R. Thank you. It was corrected.

C7. Line 47-49: with might have? Phrase structure is also strange. Rephrase please.

R. The sentence was rephrased.

C8. Line 48-52: if the information is highlighted it should be referenced.

R. A similar information was enclosed with a reference in the discussion, therefore we decided to delete the sentence from the introduction section, not to repeat information.

C9. Line 62-89: This paragraph is confusing to me. Please restructure it. Avoid alternation between histidine dipeptides and the specific amines (carnosine and anserine). Example Line 82-89, is better if connected with first sentence of line 62.

R. The layout of the text has been corrected to improve readability.

C10. Line 241 – 242: Rephrase (… which indicates that consumption of these fillets were deliver a similar amount of this bioactive compound.)

R. Thank you. The sentence was corrected.

C11. Line 390-391: remove

R. Deleted.

Reviewer 3 Report

The research is interesting. It presents solutions to problems in the industrial process by implementing alternative technologies; however, the manuscript presents some details. Abstract/instrucction

In the abstract and introduction section, it suggested that the objectives be well defined.

Line 12: it is allowed to use the same word “study” in the same sentence?

Introduction

Line 35-40: this sentence it seems not provide important information

“In the scientific literature a different 35 settings for preparing fish products using SV are proposed. Díaz et al. [2] cooked salmon 36 fillets at 80°C for 45 min, Espinoza et al. [6] heated seabream at 65°C for 46 min, whereas 37 González-Fandos et al. [3] used 90°C for 1.0 min and 3.3 min. At the same time, in the 38 internet guidelines for preparing fish SV products it was indicated that appropriate set-39 ting for the treatments are about 50°C applied for 40 to 60 min.”

Line 40-41: which temperature is considered as high? It is 60ºC considered in the “high range”?

Line 62-64: missing reference

Line 71-73: missing reference

Line 80-82: when you say “something has been demonstrated”, you should used at least one reference.

Line 83: I recommend change the word “end”, since this one is not adjusted to the context

Line 92-93-95: Word “therefore” is repeated.

Material and methods.

In line 113, indicate the vacuum pressure to which the samples were subjected (SV56, SV75, and SV 90)

A scheme would help to better understand the experimental desing.

Line 127: ash content repeated

Results

Fatty acid proportion and concentration

Line 267-277: this description of the table it seems not necessary to be included, could be deleted or resumed with a highlight of more important or noticeable result (it means, maybe something unexpected).

Line 283, explain which fatty acids are referred to with a value lower than 1 since according to table 3 C18:3, c20:5 have values higher than 1.0.

. Sensory quality

Line 353, your explanation of the values obtained seasonally is entering. It suggested to apply some texture analysis (TPA) and color analysis (CIELAB*), for quality discussion.

Microbial analyses

Table 4 continues to include the allowable limits (CFU/g) of Bacteria species, according to current regulations

Table 3: the column of P value could be deleted and explained in the description below.

Microbial quality

Table 4: could be the raw column expressed with the same power

Conclusion.

It suggested describing what is meant by "safe" since the microbiological study  not carried out during the shelf life. 

Author Response

Dear Editor and Reviewers,

Thank you for valuable comments and suggestions and the possibility to improve our paper. Changes in text were marked with a changes tracking mode. Below we enclosed the responses (R) for each Reviewers comments (C). 

 Reviewer 3

General comment: The research is interesting. It presents solutions to problems in the industrial process by implementing alternative technologies; however, the manuscript presents some details.

R. Thank you. We read your comments carefully, answered than point by point and make changes in the manuscript. All changes in the text have been marked.

C1. Abstract/instruction

In the abstract and introduction section, it suggested that the objectives be well defined.

R. The objectiveness were more detailed defined in the abstract and the end of Introduction section.

 C2. Line 12: it is allowed to use the same word “study” in the same sentence?

R. he second “study” was replaced by “investigate”. Thank you.

Introduction

C3. Line 35-40: this sentence it seems not provide important information “In the scientific literature a different settings for preparing fish products using SV are proposed. Díaz et al. [2] cooked salmon 36 fillets at 80°C for 45 min, Espinoza et al. [6] heated seabream at 65°C for 46 min, whereas  González-Fandos et al. [3] used 90°C for 1.0 min and 3.3 min. At the same time, in the internet guidelines for preparing fish SV products it was indicated that appropriate set-39 ting for the treatments are about 50°C applied for 40 to 60 min.”

R. We would like to show that different time and temperature settings are used in the studies for fish preparation and there is no strict defined one recommendation. To make it more suitable and clear the paragraph was modified.

C4. Line 40-41: which temperature is considered as high? It is 60ºC considered in the “high range”?

R. The sentence was modified. We meant that too high temperature – higher than 80 ºC might have an adverse effect on fillets quality.

C5. Line 62-64: missing reference

R. The reference was provided

C6. Line 71-73: missing reference

R. The reference was provided

C7.Line 80-82: when you say “something has been demonstrated”, you should use at least one reference.

R. The reference was provided

C8. Line 83: I recommend change the word “end”, since this one is not adjusted to the context

R. The word “end” was deleted

C9. Line 92-93-95: Word “therefore” is repeated.

R. The sentence was corrected.

Material and methods.

C10. In line 113, indicate the vacuum pressure to which the samples were subjected (SV56, SV75, and SV 90)

R. Indicated in the text

C11. A scheme would help to better understand the experimental design.

R. A scheme was added.

 C12. Line 127: ash content repeated

R. We deleted the additional sentence.

Results

Fatty acid proportion and concentration

C13. Line 267-277: this description of the table it seems not necessary to be included, could be deleted or resumed with a highlight of more important or noticeable result (it means, maybe something unexpected).

R. Thank you for the comment. The text was deleted and some information has been presented In relation to the results of Jankowska et al. as a part of discussion.

C14. Line 283, explain which fatty acids are referred to with a value lower than 1 since according to table 3 C18:3, c20:5 have values higher than 1.0.

R. The sentence in line 283 refers to n-6/n-3 ratio, which was lower than 1 as shown in Table 3.

Sensory quality

C15. Line 353, your explanation of the values obtained seasonally is entering. It suggested to apply some texture analysis (TPA) and color analysis (CIELAB*), for quality discussion.

R. Thank you for the comment. Unfortunately we are not able to repeat the experiment and to determine CIELab and TPA parameters on those samples. However, the comment is very helpful for us and we will bare it in mind when designing our future studies. We would rather to keep the discussion as it is. Since there is a correlation between sensory and instrumentally assessed texture and colour the observed changes might be explained by the differences which occur in the muscle tissue of fish, therefore in our opinion the discussion is appropriate.

Microbial analyses

C16. Table 4 continues to include the allowable limits (CFU/g) of Bacteria species, according to current regulations

R. We agree with the Reviewer that comparing results obtained with a limits would be very helpful and informative. However, according to European Union regulation to which we are familiar with, there are no limits and regulations for this kind of products, except for Listeria monocytogenes in regards to ready-to-eat products in which the growth of the bacteria is possible, which is 100 CFU/g. Therefore, we decided not to change the Table but to extend the discussion indicating this limit.

C17. Table 3: the column of P value could be deleted and explained in the description below.

R. The column with P was deleted.

Microbial quality

C18. Table 4: could be the raw column expressed with the same power

R. We are very sorry but the comment seems unclear. In the column the bacterial numbers were express in single-digit number before comma and 10 to different power which frequently used system. Therefore, we would like to keep it like this.

Conclusion.

C19. It suggested describing what is meant by "safe" since the microbiological study  not carried out during the shelf life.

R. The word “safe” was deleted from the first sentence of the conclusion to make it more clear.

Round 2

Reviewer 2 Report

All the responses were satisfactory.